# Hybridized Intelligent Home Renewable Energy Management System for Smart Grids

**Yonghong Ma [1] and Baixuan Li [1,2,*]**

[1]   School of Economics and Management, Harbin Engineering University, Harbin 150001, China;
    mayonghong0324@163.com

[2]   Academy of Agricultural Planning and Engineering, Beijing 100125, China

*   Correspondence: libaixuan@yeah.net

**Abstract:** The incorporation of renewable energies and power storage at distribution facilities are one of the important features in the smart grid. In this paper, a hybridized intelligent home renewable energy management system (HIHREM) that combines solar energy and energy storage services with the smart home is planned based on the demand response and time of consumption pricing is applied to programs that offer discounts to consumers that reduce their energy consumption during high demand periods. The system is designed and handled with minimal energy requirements at home through installation of renewable energy, preparation, and arrangement of power stream during peak and off-peak periods. The best energy utilization of residential buildings with various overlapping purposes is one of the most difficult issues correlated with the implementation of intelligent micro-network systems. A major component of the smart grid, the domestic energy control system (HIHREM) provides many benefits, such as power bill reductions, reduction in wind generation, and demand compliance. This showed that the proposed energy scheduling method minimizes the energy consumption by 48% and maximizes the renewable energy consumed at the rate 65% of the total energy generated. A new model for smart homes with renewable energies is introduced in this report. The proposed HIHREM method achieves high performance and reduces cost-utility.

**Keywords:** renewable energy management; smart home energy management

## 1. Introduction and Background of Renewable Home Energy Management

The implementation of the intelligent grid into the home means it is a smart meter, logical appliance, and tool [1]. It has to include the power management system and electricity management services throughout the energy management network [2]. The paper describes a standard model for a smart grid domestic energy distribution system that provides digital home-based energy control service consultancy [3]. The current and future intelligent grids play a significant role in delivering energy effectively, safely, and securely from sources to production, residential areas [4]. The growing demand for energy, as well as traditional sources of energy, based primarily on fossil fuels, should be fulfilled by renewable energy systems (RES) sources such as wind power, solar, and fuel cells, etc. [5]. The intelligent grid idea refers back to the electric grid coupled with an infrastructure that can allow effective, secure, reliable, and safe use of electric energy [6]. The hybridized intelligent home renewable energy management (HIHREM) network, a key component of the intelligent grid, offers several benefits, including reductions on the power bill, reducing demand and fulfilling market-side specifications [7]. The developed HIHREM system is suitable for all intelligent home environments with/without RES [8]. Figure 1 shows the home energy management system.

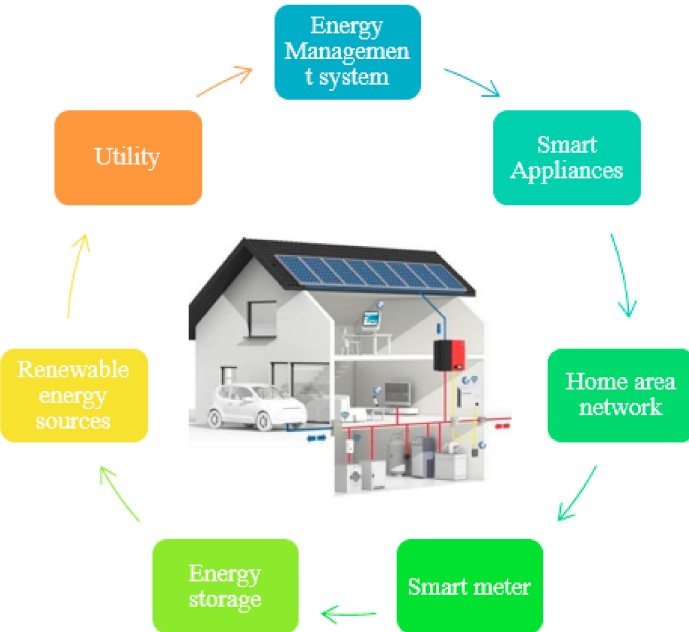

**Figure 1.** Home energy management system.

An intelligent home energy management system allows residences and suppliers to share commands to maximize their power consumption [9]. This kind of partnership between owners of energy lowers the power charges of customers and better controls renewable energy's peak charges [10]. Smart Grid is a modern R&D paradigm that combines a conventional grid with advanced IT technology which advances to increase the performance of electricity generation, transmission, delivery, and utility systems [11]. The incorporation of renewable and storage energy resources on the demand side is one key element of the intelligent grid [12]. Another important feature is that customers and the utility have the duty of controlling energy flows and use in general between themselves [13]. Smart grids have been utilized for this kind in micro/macro scales, the inefficient use of electricity would not only be minimized for households and entrepreneurs, but sources of energy would also be further optimized. In the case of smart micro grids, the use of advanced sparing power systems and integrated energy storage components can allow electronic communication between the utility and common household equipment in two ways [14]. This can provide consumers with energy efficiency resources and lets them engage in programs such as time-of-day price reduction. As buildings make a major contribution to total energy consumption, a large number of researchers worldwide have studied the issue [15]. As residences make a big contribution to the total electricity use, several researchers worldwide have studied the issue of power management on both indoor and residential power planners and suggested various power planning schemes [16]. The proposed solution would make it possible for the utility to forecast and customize the energy consumption in several housing units in a certain group:

(a) Supplying consumers with sufficient benefits (e.g., differential or TOU pricing).
(b) To ease the market through the preparation and control of appliances.

The residential customers are offered the following advantages:

i. Enhanced electricity use energy efficiencies that contributes to expenditure.
ii. Optimum local use of solar power by moving the function of some devices into the periods of usable solar energy.
iii. Full consumer satisfaction by user interaction, usage patterns, and climate.
iv. Active consumer awareness and engagement—Data on regular, weekly, and monthly trends of electricity usage and guidance on energy conservation can be given to users to meet their monthly energy requirements.

The remainder of the paper is discussed as follows: Sections 1 and 2 discussed the introduction and existing methods of smart home energy management systems. In Section 3, the hybridized intelligent home renewable energy management system has been proposed. In Section 4, the experimental results have been discussed. Finally, Section 5 concludes the research article.

## 2. Related Works and Significance of this Research Paper

Processes such as storage, supply, electric power generation, and delivery technologies can be utilized in an effective smart grid. [17]. The intelligent grid improves electricity consumption. Advanced metering infrastructure (AMI) can be used for demand estimation of a particular area if domestic devices are provided with detectors [18]. Efficient energy use is socially and economically advantageous for us [19]. While the authors implemented the period as a convenience metric for the user, they could not model the actions of various home equipment [20]. In this paper, a hybridized intelligent home renewable energy management system (HIHREM) that combines solar energy and energy storage services with the smart home is planned, tested, and implemented.

Mondal et al. [21] has discussed a multileader-multifollower Stackelberg game-theoretical pattern—a multistage and multistory game—is the issue of the distributed domestic energy-management system with capacity in an alliance of multimicro grids and several customers. To maximize their profit, Micro grid, who acts as its leader, has to decide on the minimal amount of energy to be produced by a central energy management unit. The consumers that are acting as supporters, on the other hand, need to determine the optimal energy consumption along with the energy required for stocking.

Phanichavali et al. [22] responded to variable energy prices and supply-side management which promotes smart grid users to adjust their energy consumption. To answer the differing power prices, every user in the network must consider the optimum start and running time for the devices. They proposed a greedy iterative algorithm (GIA) for each user, the expense feature and software have been designed for restrictions. Approximately gullible, the algorithmic method can be used to schedule devices for each person. They added a penalty word in the price feature for users to communicate with each other. The penalty showed significant changes in the preparation of the respective implementations. Numerical simulations demonstrated that our optimized strategy may minimize energy prices, decrease service production costs, reduce peak loads, and lower load volatility.

Logenthiran et al. [23] explained that intelligent grids research is a viable way to use Multi-Agent Network to incorporate into the energy grid that is a shared information technology. This involves many collaborative smart entities in an ecosystem. Introduction of multi-agent system (MAS) techniques are discussed in this paper for performance enhancement and power management of intelligent homes. In an intelligent building, intelligent equipments are based on agents and optimization techniques which are used in client policy-making. Both organizations are working together to reduce power usage while finding a balance between convenience, energy costs, and energy efficiency savings in the supply grid. This has contributed to the design and development of a home energy management (HEM) with MAS. This work helps intelligent homes connect and engage with and negotiate over energy sources and appliances that achieve total energy output and lowest energy charges in intelligent homes.

Tischer et al. [8] proposed that smart homes are fitted with a fuel cell to cogenerate heat and power. The PV system, an electric vehicle, a battery, and thermal power storage devices can be used to provide an integrated power management system (EMS). The EMS is based on complex forecasting and takes into account the financial implications for electricity consumption and production and the availability of the electric car, under the drivers' behaviors and priorities. Using mathematical modeling, they measured the efficiency of the proposed EMS and compared it to a simplified management system that is supposed to produce as much electricity as necessary in the home based on smart grid IoT architecture [24,25]. The results demonstrate that the approach proposed enables demand and household generation to adapt to the electricity supply conditions under a Fog-based energy management system.

The following section discusses the proposed hybridized intelligent home renewable energy management system. The cost-utility and energy demand, as well as power consumption can be expressed in the form of the mathematical model. Bidirectional communication and efficient home automation facilitate smart grid design and allow an intelligent home energy management system.

## 3. Hybridized Intelligent Home Renewable Energy Management System Overview

According to its smart network paradigm, AMI devices permit reliable two-way communication of power services in homes. This offers the additional incentive for smart houses to handle the demand side capital. This can help in managing increased energy prices by changing their energy use during power demand. Economic stimulus involves cutting electricity bills, increased efficiency of household appliance, and maintaining residential power. The HIHREM is thus defined as the best energy management system to efficiently monitor and control power sources, their processing, and usage in intelligent homes. The home area network (HANs) communications and sensing technologies can be used to gather information for power consumption from all home appliances and to track the various operating types of smart home appliances remotely in actual-time and even by personal computers or smart phones. Moreover, HIHREM offers both power storage and management services for networked energy resources (DER) and hybrid energy storage system (HESS), which are not only the ideal use of status for domestic appliances.

Figure 2 demonstrates the layout of HIHREM. The HIHREM center has a hierarchical intelligent checker that provides homeowners with tracking tools and monitoring functions based on the domestic communication system. Information for domestic electricity consumption includes scheduling and nonplanning tools, and the smart HIHREM board for the optimal shipping of demand information for household appliances. The home portal, such as the smart meter, is a networking platform for real-life installations between energy providers and the smart home. The intelligent meter usually receives a power services demand response signal as an input into the intelligent HIHREM and can be introduced to automate domestically designed devices for the home smart grids. A special type of scheduling load is electric vehicles. In addition to consuming energy from power grids to fulfill the users transport needs, it also generates backup power for other domestic cargo within the intelligent community. At present, solar image voltaic is the most specific component for distributed renewable in populated areas.

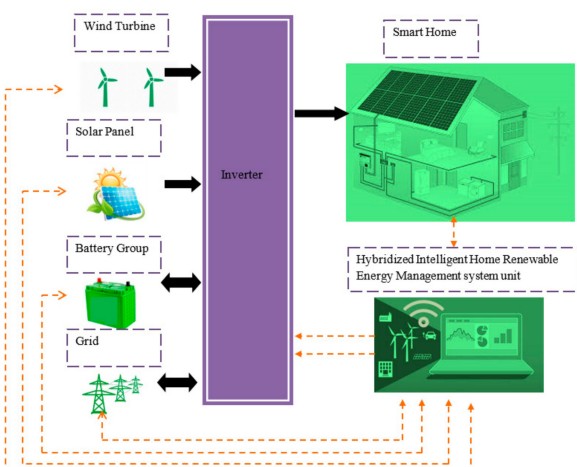

**Figure 2.** Overall architecture of hybridized intelligent home renewable energy management (HIHREM).

### 3.1. Solar Photo Voltaic (PV) and Battery Energy Storage System (BESS) are Incorporated

This modulation is intended to test the effects of renewable energies using the ToU electricity tariff. Further, the experiment is done with the modeler system Game theory based advanced metering system (GAMS), which provides numerical programming and optimization software. Hence, the power cells

are considered to be dispatched based on the machines with the capacity to absorb/distribute electricity as shown in the following limits:

$$\text{Energy limitation for output}: \ \left|P_t^E\right| \leq P_{max}^E \tag{1}$$

$$\text{Equation of load}: \ S(i+1) = S(i) - d_T P_t^E \tag{2}$$

$$\text{Equation of discharge}: \ S(i+1) = S(i) - \left(d_T P_t^E\right)/\eta^E \tag{3}$$

$$\text{Start/end of limits}: \ S(0) = S_S, S(T) = S_E \tag{4}$$

$$\text{Power storage limitation}: \ S_{min} \leq S(t) \leq S_{max} \tag{5}$$

where $P_t^E$ is the power of output over time t; max $P_{max}^E$ is a limit of maximum load/unload; $\eta^E$ is the efficiency of dumping; dT shall be the length of every time span; S($i$) is the power that is stored in the ESS before $i$; SS and SE start and finishing energy, respectively; the maximum and minimum permissible energy in ESS is Smax and Smin, respectively. The target of the lowest overall cost of electricity is as follows:

$$\text{Total\_Elect\_Cost}_{min} \ = \ \sum_{t=1}^{T} \text{TOU\_price}_t * \text{P}_{grid,t} \tag{6}$$

Additionally, the energy balance constraint is:

$$\sum_{i=1}^{I} P_{i,t} = P_t^D \quad \forall t = 1, \ldots, \ T \tag{7}$$

If $\text{P}_{grid,t}$ is the upward power grid at t; $P_{i,t}$ is the electricity request at t; $P_t^D$ is the overall demand.

Equation (8) represents the cost feature of power, while EP is the electricity price of h, K is a standardized installation and servicing price of the PV system, and PG represents the amount of electricity purchased from the grid at h.

$$\text{Electricity cos}\,t \ = \ \sum_{h=1}^{24}\left(E_P(h) * P_G(h)\right) + K(h)) \tag{8}$$

Equation (8) shows the house power consumption, where PGrid(h) indicates the grid power, Pet(h) shows the total output of electrical devices. PBattery negative value is the condition in which the equipment runs on a battery, and PBattery positive is the charge of the battery by the grid.

$$P_{Grid}(h) = P_{et}(h) - P_{PV \ panel}(h) \pm P_{Battery}(h) \tag{9}$$

To maximize the cost of electricity usage, PGrid must be lowered as minimally as possible. The proposed algorithm thus limits the maximum grid power that Equation (10) shows the limiting/restricting of the grid.

$$\text{gbm}(h) \ \geq \ \sum P_e(h) \tag{10}$$

*3.2. Optimization Requires the Following Constraints*

1.  All operating vector elements are binary.

$$x_{Wa}^t, x_{PVa}^t, x_{Ga}^t \in \{0, 1\} \tag{11}$$

2.  When the demand for the load is greater than zero at any moment, only one power source can be used to fulfill load demand. Otherwise, none of the energy sources available will be used.

$$x_{Wa}^t, x_{PVa}^t, x_{Ga}^t \leq \ 1 \tag{12}$$

3.  At any time, the wind power used by all devices at that time cannot surpass the wind power produced.

$$0 \le \sum_{a \in A} \gamma_a x_{Wa}^t \le E_W^t \tag{13}$$

4.  Solar energy consumed by all equipment cannot be greater than the solar power generated at any time.

$$0 \le \sum_{a \in A} \gamma_a x_{PVa}^t \le E_{PV}^t \tag{14}$$

5.  Energy from the grid is always used to satisfy the unfulfilled load of all equipment.

$$0 \le \sum_{a \in A} \gamma_a x_{Ga}^t \tag{15}$$

6.  The customer decides the overall working time Oh for these systems for the planned machines and their required times [αa,βa] for the operation of these appliances where αa ≤ βa and αa,βa ϵ B.

$$\sum_{t=\alpha_a}^{\beta_a} (x_{Wa} + x_{PVa} + x_{Ga}) = Oh \qquad \forall a \in R \tag{16}$$

### 3.3. Prediction of Renewable Energy

In the rise of power, a common simulation model was used to turn a National Weather Service (NWS) weather forecast into Sun or wind energy spectrum prediction. The main source of renewable in the home systems is solar energy, even though wind energy has been used as a predictive model. To sum up, shortly the model below, this forecasts solar energy harvesting using the expected sky conditions—as cloud coverage of between 0% and 100%. In addition to other climate reports, the NWS publishes a report of sky conditions at every 24 h. At all times in t, the power collection of the solar panel PS(t) has been calculated based on the sky conditions (C(t) percentage), as follows:

$$P_S(t) = P_{max} \cdot (1 - C(t)) \tag{17}$$

where Pmax is the full production power of the solar array. Therefore, the production of solar energy has been predicted in the next 24 h as follows, based on Equation (17) in any case.

$$\hat{E}_S(m+1) = \int_{kT}^{(k+1)T} P_S(\tau) d\tau \tag{18}$$

where T is the same for 24 h. To run our algorithm at the beginning of the 9 pm low rate era instead of at midnight t = 0, let's assume that t = kT, without losing generality. To use $\hat{E}_S(m+1)$ for simplicity in representing $\hat{E}_S((m+1)T)$. Re-writing Equation (18):

$$\hat{E}_S(m+1) = \int_{mT}^{(m+1)T} P_S(\tau) d\tau \tag{19}$$

### 3.4. Prediction of Energy Consumption

Exponentially weighted moving average (EWMA) model was used to predict the energy consumption of the room. The EWMA takes advantage of the diurnal aspects of household consumption and adapts them to seasonal changes. This model is very useful to foresee the comparative variations in overall energy use consumption on a normal day, with minor climatic variations, e.g., a mild day, without A/C, or daily activities, e.g., on laundry day, use of clothes dryer. More advanced models can take into account evolving activity levels, weather, or other details on weekends. One of the

purposes of this paper is to calculate how much costs can be cut with a simple and easy statistical model. Let EC(m) indicate the quantity of energy used for the kth day and $\hat{E}_C(m+1)$ indicate the expected (m + 1)th day power consumed.

$$\hat{E}_C(m+1) = \alpha \cdot \hat{E}_C(m) + (1-\alpha) \cdot E_C(m) \tag{20}$$

where $\alpha$ Factor of weighting is based on days before prediction error. As a model of TOU, pricing has different amounts of power within every single day at periodic intervals, to forecasting lower and higher power consumption in (m + 1)-days, Equations (21) and (22) can be used, respectively.

$$\hat{E}_{CL}(m+1) = \alpha \cdot \hat{E}_{CL}(m) + (1-\alpha) \cdot E_{CL}(m) \tag{21}$$

$$\hat{E}_{CH}(m+1) = \alpha \cdot \hat{E}_{CH}(m) + (1-\alpha) \cdot E_{CH}(m) \tag{22}$$

where ECL(m) and ECH(m) are the lower and higher energy consumption on the mth day, respectively.

Ultimately, the low and high amount is investigated compared to the power conversion ability of the battery charge output and the performance of the grid-tie inverter. Our aim is to charge the battery at the low power rates and to discharge the battery at a high rate to power the house. If the efficiency of power conversion is lower than the low rate to high-value ratio, battery energy storage wastes more energy in a low rate period than the direct use from the grid during a high-quota period.

### 3.5. An Effective Algorithm for Control

Using the simple prediction model of harvesting and consumption, a simple control algorithm to reduce the cost of grid power in DG installations is proposed. This will help determine how much power can be stored in the battery based on available energy, the projected weather, and the expected next 24 h electricity consumption. The control algorithm pseudo-code is displayed and described in Algorithm 1. The estimated energy within the battery that may be used on the (m+1)-th day is $\hat{E}_r(m+1)$. To measure the following $\hat{E}_r(m+1)$:

$$\hat{E}_r(m+1) = \eta \cdot E_r(m) \tag{23}$$

where h is the inverter and Er(m) capacity, the residual power inside the battery is at the start of the lower mth day rate period. In total, for every rate cycle within every day, our control algorithm encompasses the following three instances (Figure 3).

---

**Algorithm 1:** Efficient Control Algorithm

---

If $\hat{E}_r(m+1) + \hat{E}_S(m+1) \geq \hat{E}_{CH}(m+1) + \hat{E}_{CL}(m+1)$
Then,
Use the battery directly to power the building;
Else if $\hat{E}_r(m+1) + \hat{E}_S(m+1) \geq \hat{E}_{CH}(m+1)$ then,
While $\hat{E}_r(m+1) + \hat{E}_S(m+1) - \hat{E}_{CH}(m+1) > 0$ *do*
Use the battery directly to power the building;
Else if $\hat{E}_r(m+1) + \hat{E}_S(m+1) < \hat{E}_{CH}(m+1)$ *then*
While $\hat{E}_r(m+1) < \hat{E}_{CH}(m+1) - \hat{E}_S(m+1)$ do
Charge the battery;

---

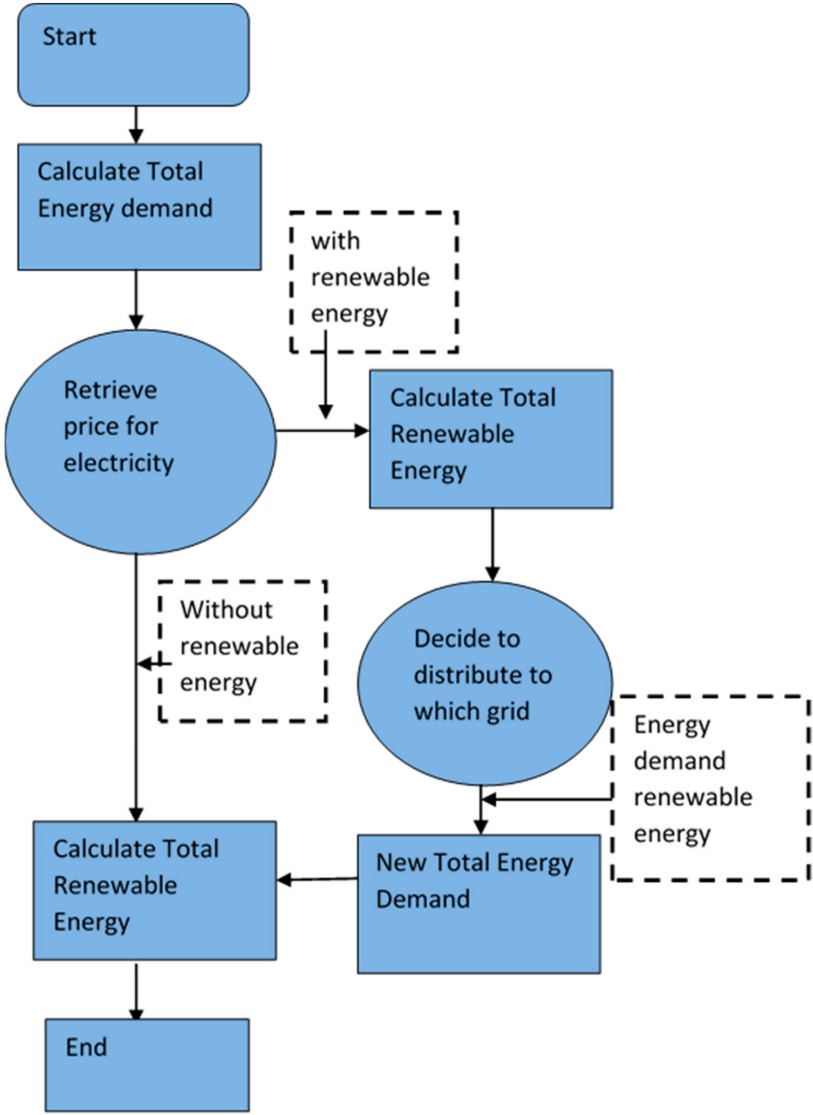

**Figure 3.** The proposed algorithm workflow.

As inferred from algorithm.1, various steps are listed as follows:

Step 1: The house does not need to use any grid electricity if the volume of battery stays estimated $\hat{E}_r(m+1)$ and solar power $\hat{E}_s(m+1)$ is greater than or equal to the projected total energy consumption in both weak and heavy cycles of energy use. The control center is used instead of powering the house directly using the power stored in the battery that includes the purchased solar power. This is unclear because our solar spectrum is not too wide to power the building, and the rest of the production takes place over a high rate era. Figure 3 shows the proposed algorithm workflow.

Step 2: Whether the battery balance is more than or equal to a high-rate energy consumption $\hat{E}_{CH}(m+1)$ cycle, and the constant solar energy $\hat{E}_s(m+1)$, then the battery shall have extra power after the time is over. The battery shall be loaded with a storage energy supply, i.e., $\hat{E}_r(m+1) + \hat{E}_S(m+1) - \hat{E}_{CH}(m+1)$. The core is running the house at a low rate using extra energy. Note that the house uses battery power only if the house is powered at a high rate during the low-rate cycle.

Step 3: If for the high-quality duration of $\hat{E}_{CH}(m+1)$, the amount of energy that the battery is anticipating remains $\hat{E}_r(m+1)$ and the projected solar power $\hat{E}_s(m+1)$, until the charging capacity and the projected solar energy is equal to the expected performance of the battery, the control center will charge the battery for the low rate, i.e., $\hat{E}_r(m+1) + \hat{E}_S(m+1) = \hat{E}_{CH}(m+1)$.

The following Section 4 shows the results. The customer is benefitted through this by reduced costs, improved services, and increased convenience. Results showed that high loads were lowered, the electricity bill is reduced, and greenhouse gas (GHG) emissions reduced. In this article, the smart appliances in the intelligent home run automatically which provides consumers with a more convenient automated fashion and higher comfort. For the implementation of a more secure, effective, and user-friendly hybridized intelligent home renewable energy management system the future Smart Grid network can be expanded.

## 4. Numerical Analysis and Its Importance

### 4.1. Efficiency Ratio Analysis

The development of a modern power grid that facilitates two-way communication between energy suppliers and customers for fine-grain calculation, control, and feedback is becoming increasingly important worldwide. Improved energy efficiency and management of available resources are some of the key features of the Smart Grid. The creation of the smart grid, a modern power grid that facilitates two-way communication between energy providers and customers for advanced metering and its control, and feedback is becoming more increasing worldwide. Increased energy efficiency and utilization of existing resources is a key feature of the smart grid. The proposed HIHREM system has a high-efficiency ratio compared to other existing methods (Table 1). Figure 4 demonstrates the efficiency ratio analysis of the proposed HIHREM system.

**Table 1.** Efficiency ratio analysis.

| No of Available Dataset | MLMF-SGT | GIA | MLN | EMS | HIHREM |
|---|---|---|---|---|---|
| 10 | 45.2 | 49.7 | 63.4 | 76.3 | 87.5 |
| 20 | 46.8 | 50.2 | 66.1 | 78.2 | 90.3 |
| 30 | 47.9 | 54.4 | 67.5 | 74.5 | 87.7 |
| 40 | 40.2 | 49.8 | 60.3 | 70.2 | 96.5 |
| 50 | 39.3 | 48.5 | 58.7 | 69.1 | 97.3 |

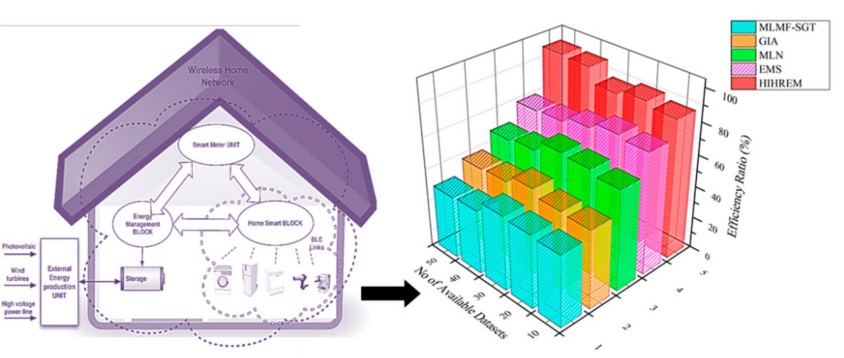

**Figure 4.** The efficiency ratio of HIHREM system.

### 4.2. Cumulative Cost-Utility Evaluation

Every appliance has been initialized to start at its optimal start time and work in its ideal modes before the optimization algorithm starts. When the changes in user cost between successive iterations fall below a limit for all users, the iteration of the distributed algorithm is terminated. As a consequence of the concurrent changes, in this situation the cost cannot be penalized and the penalty ratio decreases with the company's generation cost index based on the prediction ratio. When using the proposed HIHREM system the cost-utility can be minimized when compared to other existing methods. Figure 5 shows the cumulative cost-utility evaluation.

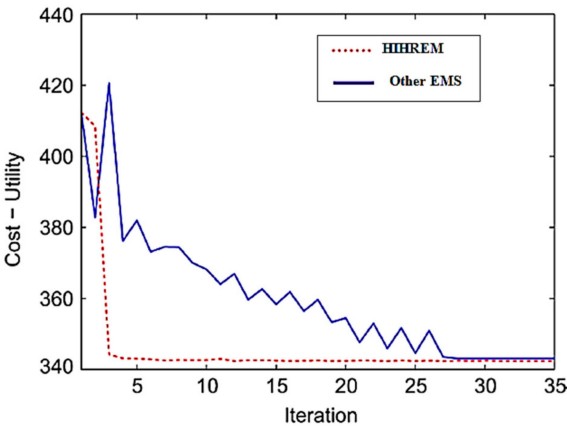

**Figure 5.** Cumulative cost-utility evaluation.

### 4.3. Power Consumption Rate

Demand response (DR) is a key smart grid technology. The response to demand can be viewed since the changes in electricity use is by end-use consumers. Increased electric energy prices over time, changes in the incentive packages designed to induce lower electricity usage at high market prices or at a period when system efficiency is compromised may be the reason for that. However, an automated architecture that monitors and adapts dynamically to real-time information could be the trend for consumers to implement certain DR strategies manually. The proposed HIHREM method has less power consumption rate when compared to other existing MLMF-SGT, GIA, MLN, and EMS methods. Figure 6 shows the power consumption rate in an intelligent home environment.

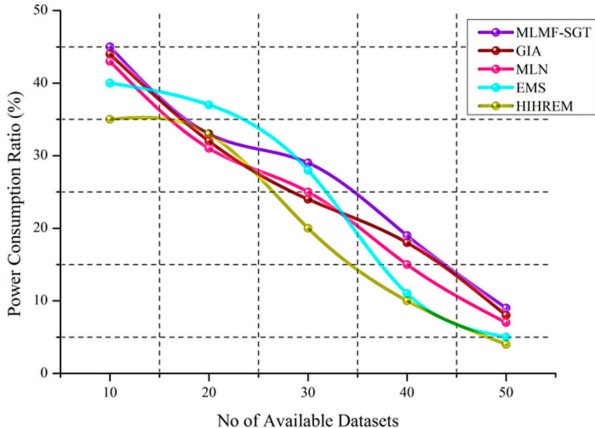

**Figure 6.** Power consumption ratio.

Table 2 shows the power consumption ratio of the proposed HIHREM method. Research in the field of home energy management found that it could be composed of energy consumption scheduling or domestic grids. The SLP (service location protocol) time series system is utilized to see energy use in individual households. This technique produces a real-time series of typical electricity demand load profiles of up to 1 s high resolution.

**Table 2.** Power consumption ratio.

| No of Available Dataset | MLMF-SGT | GIA | MLN | EMS | HIHREM |
|---|---|---|---|---|---|
| 10 | 45.2 | 44.7 | 43.4 | 40.3 | 35.5 |
| 20 | 33.8 | 32.2 | 31.1 | 37.2 | 30.3 |
| 30 | 29.9 | 24.4 | 25.5 | 28.5 | 20.7 |
| 40 | 19.2 | 18.8 | 15.3 | 11.2 | 10.5 |
| 50 | 9.3 | 8.5 | 7.7 | 5.1 | 4.3 |

### 4.4. Overall Performance Ratio

The proposed HIHREM system performance is tested by two examples. In the first example, the residents of the home are expected to leave from 7:00 AM. They are then working until noon. The average active resident of the home is 36.72%, 31.15%, and 32.13% away. The predefined schedule of operation for both scenarios has been explained. The overall performance of the intelligent home energy management system is high. Overall optimal behavior is enforced through the price vector, an invisible man that coordinates the interaction between users, and a penalty term that penalizes significant changes to user timetables between iterations enhance the convergence of the algorithm. The performance of the proposed algorithm is confirmed by numerical simulations. Figure 7 shows the overall performance of the proposed HIHREM system.

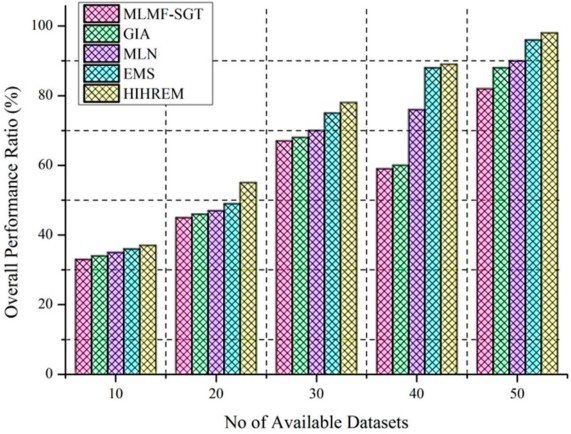

**Figure 7.** Overall performance ratio.

### 4.5. Energy Demand per Hour

The steep rise in demand for electricity has posed a serious challenge for the electricity distribution networks, with most utilities adopting a load shedding trend; it represents a method of handling load requirements by shedding them in critical situations where demand is higher than a total generation to avoid system failure or major collapse. Load management is a demand-side management system run by a consumer power supply or energy management system. In the implementation of the residential demand response (DR) programs in the smart environment, home energy management systems (HEM) play a critical role. This helps a homeowner to execute intelligent charging controls automatically based on utility systems, consumer choice, and charging priority. Figure 8 shows the energy demand per hour in an intelligent home environment.

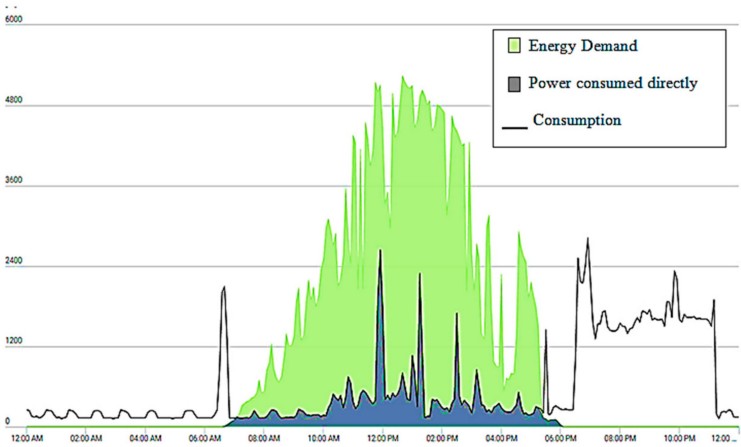

**Figure 8.** Energy demand per hour.

The usage of residential energy will differ depending on a number of factors, such as house size, number of people, location, and the season. The inter arrival times between two requests are exponentially negative, with a mean of 12 h. These parameters affect the household's heating, cooling, lighting, and related load. The inter arrival time is exponentially distributed with a mean of 2 h during morning peak periods and evening peak periods. The power consumption and energy demand will increase while the number of appliances increases.

## 5. Conclusions

This paper presents the hybridized intelligent home renewable energy management system (HIHREM). The mechanism proposed is intended to minimize the cost of smart home electricity by maximizing renewable energy use. In the model, it is assumed that the energy consumption of all appliances is constant at every interval and it is not constant every time. This showed that the proposed energy scheduling method minimizes the energy consumption by 48% and maximizes the renewable energy consumed at the rate 65% of the total energy generated. The proposed efficient control algorithm reduces the complexity and controls home energy consumption. Demand management systems are a very effective way to control consumer's electricity resources properly. This not only minimizes bills or saves energy but also will increase the efficiency of power grids by moving the load to off-peak hours, by adjusting demand to supply ratio of renewable energy or by its way of reaction to emergency conditions. To control the electric energy of residential customers, two-time scales can be used which are the day ahead and real-time. In the day-to-day case, the user operating plan (or generic future time horizon) is established based on data forecasting over the next 24-h cycle. In all the above respects, the proposed HIHREM method has high performance when compared to other existing methods.

**Author Contributions:** Y.M. conceived the idea, designed research process, and drafted the manuscript; B.L. performed the experiments, produced the results, contributed to translate and modify the paper. All authors have read and agreed to the published version of the manuscript.

**Funding:** This work funded by the National Natural Science Foundation of China (71874040).

**Acknowledgments:** The authors thank the anonymous reviewers and editors for their constructive suggestions.

**Conflicts of Interest:** The author declares no conflict of interest.

## Nomenclature

| | |
|---|---|
| [αa, βa] | Interval of running time for expected systems a |
| $\gamma_a$ | Appliance's nominal power |
| R | All smart home appliances (hours) set |
| $E_{PV}^t$ | Solar power generated on-time t |
| $E_W^t$ | Wind power generated on time t |
| Oh | Hours of operation |
| Xa | Pointer of appliance operating times a |
| $x_{Ga}^t$ | The use of grid power to meet appliance demand at a time |
| $x_{PVa}^t$ | The use of solar power to meet appliance demand at a time |
| $x_{Wa}^t$ | The use of wind power to meet appliance demand at a time |

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
