# Peer review of "Hybridized Intelligent Home Renewable Energy Management System for Smart Grids"

_sustainability, doi:10.3390/su12052117_

Round 1

Reviewer 1 Report

Grammatical errors throughout:

line 32 sentence doesn't make sense

Define: RES, AMI

Figure 1: Energy Management system, in the diagram the t in Management is on the next line

Line 47 doesn't make sense

lines 65-69 some start with Capital letters some start with lowercase and the justification needs to be moved to the right

line 121 the "and" and the "comma" are not right

Figure.2 should say Figure 2

No Figure # on line 283

there are several areas where an apostrophe is in the middle of a sentence and not tight against the word to pluralize

there are spacing issues and other grammatical issues throughout the article

Author Response

Grammatical errors throughout:

Ans: Checked and verified

line 32 sentence doesn't make sense

Ans: Checked and Verified

Define: RES, AMI

Ans: Resource Energy management system(RES) and advanced metering infrastructure (AMI)

Figure 1: Energy Management system, in the diagram the t in Management is on the next line

Ans: An intelligent home energy management system allows residences and suppliers to share commands to maximize their power consumption [9]. This kind of partnership between owners of energy lowers the power charges of customers and better controls renewable energy's peak charges [10].

Line 47 doesn't make sense

Ans: it is verified and corrected.

lines 65-69 some start with Capital letters some start with lowercase and the justification needs to be moved to the right

Ans: it is verified and corrected line 121 the "and" and the "comma" are not right

Figure.2 should say Figure 2

Ans: it is verified and corrected

No Figure # on line 283

Ans: it is verified and corrected

there are several areas where an apostrophe is in the middle of a sentence and not tight against the word to pluralize

Ans: it is verified and corrected

there are spacing issues and other grammatical issues throughout the article

Ans: it is verified and corrected

Reviewer 2 Report

ï‚· In the home, systems are solar energy, even though wind energy has been used as a
predictive model. To sum up, shortly the model below, which forecasts solar energy
harvesting using the expected sky conditions — as a cloud coverage of between 0% and
100%. Based on the above line, the author should define what is sky conditions, how it will be
used for solar energy harvesting and how the cloud coverage of between 0% and 100%

ï‚· The Manuscript needs to be edited for grammar and spell check required. The authors are
informed to keep images, graphs and data tables in clear
ï‚· In page no.6, the Pmax is the full production power of the solar array and How the
production of solar energy has been predicted in the next 24 hours based on equation (17)

ï‚· In equation (16) the overall working time is Oh and their required times [αa,βa] for the
operation of these appliances where αa≤βa and αa,βa ϵ B. Here what is

ï‚· Some of the Sentences are not clear in this paper check throughout the paper and English
correction needed. For instance “A special type of scheduling load is electric vehicles. In
addition to consuming energy from power grids to fulfill users ' transport needs, it
generates backup power for other domestic cargo within the intelligence community.” is
not clear

ï‚· The proposed methodology is having some difficulties behind the problem analysis and
the motivational aspect of the system needs some more enhancements.

Author Response

 In the home, systems are solar energy, even though wind energy has been used as a predictive model. To sum up, shortly the model below, which forecasts solar energy harvesting using the expected sky conditions — as a cloud coverage of between 0% and 100%. Based on the above line, the author should define what is sky conditions, how it will be used for solar energy harvesting and how the cloud coverage of between 0% and 100%

Ans: it is verified and corrected.

ï‚· The Manuscript needs to be edited for grammar and spell check required. The authors are informed to keep images, graphs and data tables in clear.

Ans: The grammar and spell have been corrected and the images, graphs and data tables have been corrected.

ï‚· In page no.6, the Pmax is the full production power of the solar array and How the production of solar energy has been predicted in the next 24 hours based on equation (17).

Ans: The method has been included in the paper.

ï‚· In equation (16) the overall working time is Oh and their required times [αa,βa] for the operation of these appliances where αa≤βa and αa,βa ϵ B. Here what is.

Ans: It is explained in the paper.

ï‚· Some of the Sentences are not clear in this paper check throughout the paper and English correction needed. For instance “A special type of scheduling load is electric vehicles. In addition to consuming energy from power grids to fulfill users ' transport needs, it generates backup power for other domestic cargo within the intelligence community.” is not clear.

Ans: The language has been corrected.

ï‚· The proposed methodology is having some difficulties behind the problem analysis and the motivational aspect of the system needs some more enhancements.

Ans: it is verified and corrected

Reviewer 3 Report

The findings of the article on Hybridized intelligent home renewable energy management system (HIHREM) for smart home are comprehensive. The scientific proof of the article needs to include in the abstract for readability. Authors are strongly commented to add numerical outcomes in the abstract and conclusion. It is suggested to check the logical consistency for the Eq(8) and Eq(10), Kindly Clarify with mathematical proof.
Authors need to be more specific about the usage of parameters which is not explained with definition and corresponding parameters has been mentioned below for reference:
- Prediction of renewable energy
- Prediction of power consumption
Authors are informed to abbreviate the following terms: HAN, AME,HESS, PV, GAMS, BESS. “This modulation is intended to test the effects of renewable energies and BESS for undisputed consumption of energy using the ToU electricity tariff. The experiment is done with the modeler system GAMS, which provides numerical programming and optimization software. The above line does not provide intended meaning the author should check and clarif.
The writing of the paper needs an improvement in English Correction. There is a need to mention proper justification of proposed methodology in the abstract with their observed experimental results. The following citations have to be included in the revised version of the paper:

Hadjioannou, Vasos, et al. "Security in smart grids and smart spaces for smooth IoT deployment in 5G." Internet of Things (IoT) in 5G Mobile Technologies. Springer, Cham, 2016. 371-397.

Nikoloudakis, Yannis, et al. "A fog-based emergency system for smart enhanced living environments." IEEE Cloud Computing 3.6 (2016): 54-62.

Author Response

  • The findings of the article on Hybridized intelligent home renewable energy management system (HIHREM) for smart home are comprehensive. The scientific proof of the article needs to include in the abstract for readability. Authors are strongly commented to add numerical outcomes in the abstract and conclusion. 

Ans: This showed that the proposed energy scheduling method minimizes the energy consumption by 48 % and maximizes the renewable energy consumed at the rate 65 % of the total energy generated. A new model for smart homes with renewable energies is introduced in this report. The proposed HIHREM method achieves high performance and reduces cost-utility. 

  • It is suggested to check the logical consistency for the Eq(8) and Eq(10), Kindly Clarify with mathematical proof.

Ans: It is checked and verified

  • Authors need to be more specific about the usage of parameters which is not explained with definition and corresponding parameters has been mentioned below for reference

Prediction of renewable energy

Prediction of power consumption

Ans: it has been included in the results and discussion section under the subsection (iii) and (iv)

  • Authors are informed to abbreviate the following terms: HAN, AME,HESS, PV, GAMS, BESS. “This modulation is intended to test the effects of renewable energies and BESS for undisputed consumption of energy using the ToU electricity tariff. The experiment is done with the modeler system GAMS, which provides numerical programming and optimization software. The above line does not provide intended meaning the author should check and clarify.

Ans: This modulation is intended to test the effects of renewable energies using the ToU electricity tariff. Further, the experiment is done with the modeler system GAMS, which provides numerical programming and optimization software. Hence, The power cells are considered to be dispatch based on the machines with the capacity to absorb/distribute electricity as shown in the following limits: Home area network (HAN), Advanced metering infrastructure (AMI), hybrid Energy storage system (HESS), Photo Voltaic (PV), Game theory based advanced metering system (GAMS), battery Energy storage system (BESS). 

  • The writing of the paper needs an improvement in English Correction. There is a need to mention proper justification of proposed methodology in the abstract with their observed experimental results. The following citations have to be included in the revised version of the paper:Hadjioannou, Vasos, et al. "Security in smart grids and smart spaces for smooth IoT deployment in 5G." Internet of Things (IoT) in 5G Mobile Technologies. Springer, Cham, 2016. 371-397.Nikoloudakis, Yannis, et al. "A fog-based emergency system for smart enhanced living environments." IEEE Cloud Computing6 (2016): 54-62.

Ans:The language has been corrected and the reference has been included in the paper.
